# A time series analysis of transparent exopolymer particle distributions and C:N stoichiometry in the subtropical North Pacific: a key process in net community production and preformed nitrate anomalies?

Kieran Curran[1], Tracy A. Villareal[2], Robert T. Letscher[1,3]

1   Ocean Process Analysis Laboratory, University of New Hampshire, Durham, NH, 03824 USA
2   Marine Science Institute, University of Texas at Austin, Port Aransas, TX 78373 USA
3   Department of Earth Sciences, University of New Hampshire, Durham, NH 03824 USA

*Correspondence to*: Robert T. Letscher (robert.letscher@unh.edu)

## Abstract

Within the oligotrophic subtropical oceans, summertime dissolved inorganic carbon drawdown despite nutrient limitation in surface waters and subsurface oxygen consumption in the absence of Redfieldian stoichiometric nitrate release are two phenomena still awaiting a full mechanistic characterization. Many processes may contribute to these anomalies including $N_2$ fixation, non-Redfieldian DOM cycling, vertically migrating phytoplankton, heterotrophic $NO_3^-$ uptake and vertical $NO_3^-$ injection events. While these processes have been measured or modelled they generally cannot fully account for the magnitudes of oxygen/nitrate anomalies and excess dissolved inorganic drawdown observed in many oligotrophic subtropical waters. One other candidate process that may contribute to both phenomena is the formation of carbon-rich transparent exopolymer particles (TEP) and Coomassie-stainable particles (CSP) from dissolved organic precursors in surface waters and their subsequent export and remineralization below, however, few TEP and CSP data exist from the oligotrophic ocean. Here we present a multi-year time-series (Jan 2020 – Sep 2022) analysis of TEP, CSP and total dissolved carbohydrate concentrations at both Station ALOHA (22˚45',158 ˚W) and along a meridional transect during June 2021 from 22˚45' to 31˚N within the North Pacific subtropical gyre. Exopolymer C:N stoichiometry at Station ALOHA varied between 16.4 – 34.3, with values being more carbon-rich in summer (26-34); ratios were higher (33-38) toward the gyre centre at 31˚N. TEP concentrations were consistently elevated in surface waters through Spring-Autumn (4-8 µM C after carbon conversion) at Station ALOHA with lower concentrations (~1.5-3 µM C) and more uniform vertical distribution during winter, indicating that TEP accumulated in surface waters may vertically sink and be exported with winter mixing. The accumulation of exopolymers in surface waters through Spring-Autumn and its subsequent vertical export may account for 6.5-20% of net community production, helping to reduce the estimated imbalance of N supply and demand at this site to <10%. The upper ocean exopolymer cycle may explain 22-67% of the observed oxygen/nitrate anomalies, helping to close the C, N, and $O_2$ budgets at station ALOHA, while leaving room for significant contributions from other processes such as vertically migrating phytoplankton and heterotrophic nitrate uptake. These results suggest that exopolymer production and cycling may be more important to open ocean carbon biogeochemistry and the biological pump than previously expected.

# 1 Introduction

Subtropical oceans constitute one of earth's largest biomes, where the euphotic water column exhibits sustained macronutrient limitation due to strong thermal stratification (Reygondeau et al., 2013). Consistently low euphotic zone chlorophyll concentrations observed in these regions lead to depressed primary production estimates using ocean-colour satellite and bio-optical float profile data (Longhurst et al., 1995; Long et al., 2021; Westberry et al., 2023). Despite this assumption of low productivity, various measured rates of annual net community production (NCP) and total annual carbon export from the ocean subtropics suggest a biological pump strength that is maintained at levels consistent with mesotrophic oceanic regions receiving a higher vertical nutrient injection flux (Gruber et al., 1998; Emerson, 2014; Teng et al., 2014; Roshan and DeVries, 2017; Quay et al., 2020; Karl et al., 2021; Quay and Stephens, 2025).

Moderate rates of summertime surface dissolved inorganic carbon (DIC) drawdown are observed in low-chlorophyll Atlantic and Pacific subtropical oceans (2-3 mol C $m^{-2}$ $y^{-1}$) despite limiting nitrate and phosphate concentrations, and stratification that would seem to limit diapycnal supply of nutrients to the euphotic zone for most of the year (Sambrotto et al., 1993; Michaels et al., 1994; Dave and Lozier, 2010; Williams et al., 2013; Emerson, 2014). Processes of nutrient enrichment such as $N_2$ fixation, episodic mixing events, and iron-rich dust deposition are unable to fully provide sufficient nutrient supply to sustain this persistent summertime anomaly (Johnson et al., 2010; Chow et al., 2017; Fawcett et al., 2018; Letscher and Villareal, 2018; Letelier et al., 2019; Karl et al., 2021). In addition, most subtropical regions exhibit subsurface respiration without concomitant nitrate release expected from the remineralization of Redfieldian organic matter. This produces a widespread negative preformed nitrate ($preNO_3^-$) anomaly between ~120-180m (Emerson and Hayward, 1995; Abell et al., 2005; Ascani et al., 2013; Letscher and Villareal, 2018; Smyth and Letscher, 2023), and stoichiometrically balanced positive $preNO_3^-$ anomalies found within the upper 100m where $O_2$ is produced without stoichiometric $NO_3^-$ drawdown (Letscher and Villareal, 2018). The introduction of allochthonous macronutrient supply to the surface mixed layer by vertically migrating phytoplankton or the production and export of non-Redfieldian organic matter (high elemental carbon:nitrogen ratio) are two potential processes which may couple these three phenomena and help explain the elevated surface DIC drawdown and positive $preNO_3^-$ as well as negative $preNO_3^-$ below the sub-surface chlorophyll maximum in these regions (Letscher and Villareal, 2018) .

Transparent exopolymer particles (TEP), mostly comprised of acidic polysaccharides, are ubiquitous throughout the oceans, where they tend to accumulate in surface waters due to their low density (Azetsu-Scott and Passow, 2004). Exopolymers are typically observed as being carbon-rich, with C:N ratios of >20:1 (Mari et al., 2001; Engel and Passow, 2001; Passow, 2002b; Guo et al., 2022), which makes them a candidate for surface mixed layer DIC drawdown with minimal nitrogen requirement, particularly if composed of pure carbohydrate (e.g. 1 C : 1 $O_2$ : 0 N). While most abundant during large blooms of phytoplankton in eutrophic waters, TEP and their precursors are produced by a wide variety of phytoplankton and bacteria across different marine and aquatic environments (Passow et al., 1994; Nosaka et al., 2017; Zamanillo et al., 2019). Exopolymers act as a bridge between the dissolved and particulate fractions of marine organic matter, with dynamic assembly and disassembly of marine gels helping to fill the size continuum of particles in the ocean (Verdugo et al., 2004; Verdugo, 2012). The related but distinct Coomassie stainable particles (CSP) are thought to track the more protein-rich component of the marine exopolymer/gel pool, which likely impacts the fate of these particles differently

than the polysaccharide-rich TEP pool (Cisternas-Novoa et al., 2015; Zamanillo et al., 2021). TEP
contributes to sinking exopolymer aggregates, which in turn constitute a significant flux of POC to the
upper mesopelagic zone where much of this organic matter may be consumed by aggregate-associated
bacteria (Wurl et al., 2011b; Nagata et al., 2021) and zooplankton (Ling and Alldredge, 2003).
TEP production from phytoplankton exudates is associated with excess DIC drawdown even in nutrient-
replete water. In these regions, carbon overconsumption can be as high as 30-40% with respect to nitrate
and phosphate removal and POM C:N:P stoichiometry (Toggweiler, 1993). Surface mixed layer
exopolymer production may increase as cells are stressed by nutrient limitation or photo-oxidative
stresses (Berman-Frank et al., 2007; Ortega-Retuerta et al., 2009a; Iuculano et al., 2017), persistent in
many subtropical surface waters. Therefore, despite lower phytoplankton biomass in these oligotrophic
regions, significant TEP production and seasonal variability may still occur.
Given that different oligotrophic regions exhibit significant variability in the elemental stoichiometry of
organic matter including biomass (Martiny et al., 2013), detrital POM, and DOM (Letscher and Moore,
2015; Liang et al., 2023), across depth and time, region-specific measurements are needed to quantify the
importance of exopolymer particles to pelagic biogeochemistry of different regions (McCarthy et al., 1996;
Mari et al., 2001; Passow, 2002b; Beauvais et al., 2003).
In this study, we assess whether significant depth, temporal, and latitudinal gradients exist in: 1).
exopolymer abundance and its associated C:N content that may help to explain the seasonal excess DIC
drawdown in the absence of known nutrient supply pathways (e.g. Johnson et al., 2010),  and 2)
potentially related subsurface $preNO_3^-$ anomalies present within the North Pacific subtropical gyre
(NPSG). To do this, we sampled two classes of exopolymers: carbohydrate-rich transparent exopolymer
particles (TEP) and protein-containing Coomassie-stainable particles (CSP) as well as dissolved
carbohydrates (precursor molecules of larger exopolymer particles (Passow, 2000; Ortega-Retuerta et
al., 2009b; Arnosti et al., 2021)), for nearly three years to quantify their concentrations, vertical
distributions, and seasonal and latitudinal variability.

In order to produce quantitative estimates of TEP and CSP concentrations, we also directly estimated the
organic C and N content of exopolymers spontaneously assembled under controlled conditions in the field
to convert TEP and CSP values to carbon and nitrogen equivalents. With these quantitative estimates of
TEP-C and CSP-N concentrations, we then discuss the potential contributions of the exopolymer cycle for
explaining surface mixed layer excess DIC drawdown and subsurface $preNO_3^-$ anomalies. This contribution
helps to close the C, N, and $O_2$ budgets at station ALOHA, and may apply to the carbon and nutrient
biogeochemistry of the subtropical oceans more generally.

## 114 2 Methods

### 115 2.1 Sample collection

Water samples for measurements of TEP, CSP, and dissolved polysaccharides were collected using a
Niskin rosette onboard the RV *Kilo Moana* from 15 cruises between January 2020 and September 2022.
14 cruises were part of the Hawaiian Ocean Time-series (HOT) sampling program at Station ALOHA (22°
45' N 158° W), with 1 cruise sampling 10 stations in the North Pacific Gyre along a nominal 158ºW
transect from Station ALOHA to 31° N during June 2021, also on RV *Kilo Moana* (Fig. 1). Vertical profiles
of salinity (Sea-Bird SBE-09), temperature (Sea-Bird SBE-3 Plus) and oxygen (Sea-Bird SBE-43) were also
collected from the rosette CTD instrument package. Primary productivity, chlorophyll a, particulate
carbon, and particulate nitrogen data measured as part of the HOT program for the period 1988-2022
were obtained from https://hahana.soest.hawaii.edu/hot/hot-dogs/.


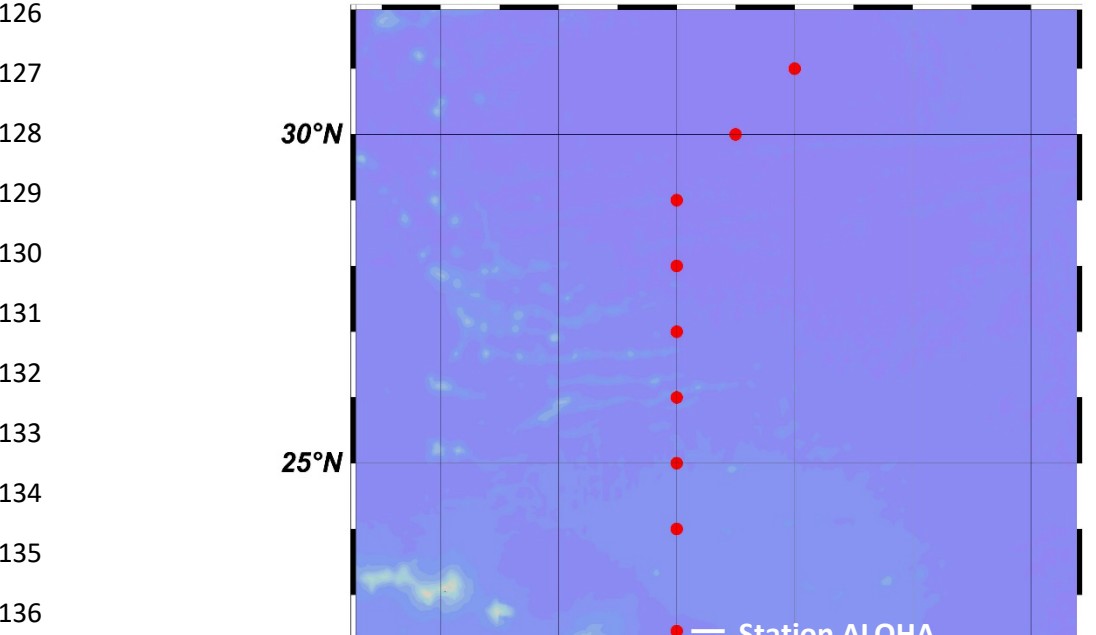

Figure 1. Map showing the location of Station ALOHA where time series measurements were collected and the
143                         stations along the June 2021 transect between ALOHA and 31°N.

2.2 Quantification of transparent exopolymer particles (TEP) and Coomassie-stainable
particles (CSP)
Water samples for TEP and CSP (0.5-2.0L) were taken from 5-8 depths and stored in polycarbonate bottles
(Corning) in blacked-out carriers until filtration. Samples were processed from deepest to shallowest to
minimize any effects of small temperature changes on exopolymer formation dynamics. Water samples
for TEP and CSP were filtered using 0.4 µm pore-size, 25mm diameter polycarbonate filters (Whatman)
using a peristaltic pump (Cole-Parmer) and silicone tubing (Masterflex). Filters were then placed onto a
vacuum filtration rig and dyed with acidified (pH 2.5) 0.02% Alcian Blue (AB) solution (Alcian Blue 8X,
Sigma Aldrich) for TEP samples following Bittar et al. (2015) and 0.04% Coomassie brilliant blue (CBB)
(SERVA) solution (pH 7.4) for CSP samples following Cisternas-Novoa et al. (2015). Dyed filters were placed
in polypropylene vials (Falcon) and frozen at -20ºC, and 2-day shipped back to the shore-based laboratory
in ice-packed coolers (Pelican). TEP samples were extracted in 6 ml 80% sulphuric acid solution for 2 hrs
and absorbance read at 787nm. CSP samples were extracted in 4 ml 3% sodium dodecyl sulphate (SDS) in
50% isopropyl alcohol solution for 2 hrs at 37°C under ultrasonication and read at 615nm. Absorbance
values were blanked against the same type of polycarbonate filters after filtration of 500ml ultrapure
water. Blanks were also taken with 500ml 0.2 µm filtered seawater to check that there was no bias
resulting from sub-0.2 µm organic material from seawater retained on the filters. These blanks were not
significantly different and had a combined coefficient of variation of 0.039.  Absorbance values were
calibrated against a dilution series of xanthan gum (XG) (Sigma) and bovine serum albumin (BSA) (Sigma)
for TEP and CSP respectively. Concentration units are therefore expressed as µg XG equivalents $L^{-1}$ and µg
BA equivalents $L^{-1}$ following the literature convention (e.g., Cisternas-Novoa et al. (2015)) using the
spectrophotometric method for TEP and CSP quantification in Figure 2. TEP sample replicates had a mean
coefficient of variation of 0.04 µg XG equiv. $L^{-1}$ and CSP samples 0.14 µg BA equiv. $L^{-1}$  (n=24) from 8 sets
of triplicate measurements.

## 168 2.3 Dissolved carbohydrates

Water samples for dissolved carbohydrate analysis were gravity filtered from the Niskin rosette using a
47mm combusted GF/F filter (Whatman; 0.7µm nominal pore size) into acid cleaned and furnaced glass
vials. Vials were frozen at -20ºC and transported similar to above for lab analysis. Using the approach of
Myklestad et al (1997), total HCl-hydrolysable carbohydrates (TCHO) were measured against a glucose
calibration standard and expressed in µM carbon. The method uses the alkaline ferricyanide reaction with
2,4,6-tripyridyl-*s*-triazine (TPTZ) that produces a deep violet color with reduced iron, allowing sensitive
measurement of low carbohydrate concentrations with spectrophotometry. Reagents were made fresh
for each run of samples and kept in blacked-out glassware. Coefficients of variation averaged 2.5% on
triplicate analyses of dissolved carbohydrate.

## 178 2.4 Carbon and Nitrogen conversion factors

During field sampling at station ALOHA (22.75°N, 158°W) and from 31° N, 156°W in June 2021 and October
2021 from station ALOHA alone, 3 x 10 litre volumes of seawater from two depths (5m, 125m) were
filtered through a 0.2 µm capsule filter (Pall) into opaque HDPE plastic bottles and stored in the dark while
at sea at sample depth temperature ±1°C. Bottles were left for 80-100 hrs to allow sufficient time for
exopolymer to spontaneously reform from the dissolved fraction. From these bottles, duplicate filtrations
(1.5L) were performed for TEP and CSP concentrations as above and duplicate filtrations for particulate
carbon and nitrogen were taken onto 47mm GF filters (Whatman) for CHN analysis of the collected
exopolymer particles.
Particulate carbon and nitrogen data (in µM C and µM N) were then used with the measurements of TEP
and CSP (in µg XG equiv. $L^{-1}$ and µg BA equiv. $L^{-1}$) to convert the latter exopolymer concentration units to
µM C and µM N using carbon and nitrogen conversion factors (CCF and NCF).

$$CCF = \frac{\text{µM } Particulate\ Carbon}{\text{µg } XG\ equiv\ \text{L}^{-1}} \quad\quad\quad\quad (1)$$
$$NCF = \frac{\text{μM } Particulate\ Nitrogen}{\text{μg } BA\ equiv\ \text{L}^{-1}} \qquad\qquad\qquad (2)$$
TEP carbon (TEP-C) and CSP nitrogen (CSP-N) concentrations are thereafter converted and expressed in
μM units of carbon and nitrogen respectively.

# 196 3 Results

## 197 3.1 Carbon and Nitrogen conversion factors

Table 1. TEP-C and CSP-N conversion factors and exopolymer C:N ratios measured from exopolymer ingrowth
incubations of 0.2 μm-filtered seawater conducted in June and October '21 at station ALOHA and at the northern
end of the June '21 transect (31°N, 156°W); values in parentheses are coefficients of variation.

| Conversions | TEP-C Jun 21 | TEP-C Oct 21 | CSP-N Jun 21 | CSP-N Oct 21 | C:N Jun 21 | C:N Oct 21 |
|---|---|---|---|---|---|---|
| | | | | | | |
| ALOHA 5 m | 0.529 (0.02) | 0.577 (0.02) | 0.018 (0.03) | 0.012 (0.18) | 25.7 (0.01) | 18.54 (0.16) |
| ALOHA 125 m | 0.627 (0.05) | 0.600 (0.19) | 0.005 (0.23) | 0.013 (0.27) | 34.3 (0.11) | 16.40 (0.36) |
| 31°N 5m | 0.656 (0.12) | | 0.004 (0.05) | | 33.2 (0.04) | |
| 31°N 125m | 0.759 (0.05) | | 0.003 (0.19) | | 38.1 (0.01) | |


Carbon conversion factors for TEP-C at station ALOHA varied between 0.529-0.627 μM C per μg XG equiv
L$^{-1}$ with mean surface values being lower than at 125 m (p = 0.07, Welch's t-test) (Table 1). These values
are consistent with the frequently used conversion factor of 0.6 from Engel and Passow (2001). Nitrogen
conversion factors for CSP-N varied by a factor of ~6 between 0.003-0.018 with lower organic nitrogen
content found at 31ºN than at station ALOHA (p = 0.04, Welch's t-test) (Table 1.). The C:N ratio (16.4-34.3)
at ALOHA varied more than carbon conversion factors (0.529-0.627), e.g. by a factor of ~2 and ~1.2
respectively, with summertime samples from 125 m being most carbon-rich and samples from October at
125 m having the lowest C:N ratios (p = 0.007, Welch's t-test). All samples were carbon-rich with respect
to the canonical Redfield ratio, with exopolymer C:N ratios at station ALOHA being significantly higher in
summer than autumn at 5 m (p = 0.025, Welch's t-test) and 125 m (p = 0.007, Welch's t-test), consistent
with the observations of (Michaels et al., 1994). Summertime C:N ratios were significantly higher in
northern gyre-associated waters (31°N) than at station ALOHA, e.g. 33 – 38 vs. 26 – 34, both for 5 m (p =
0.0001, Welch's t-test) and at 125 m (p = 0.01, Welch's t-test).


## 3.2 Interannual variation in TEP, CSP, and TCHO at station ALOHA

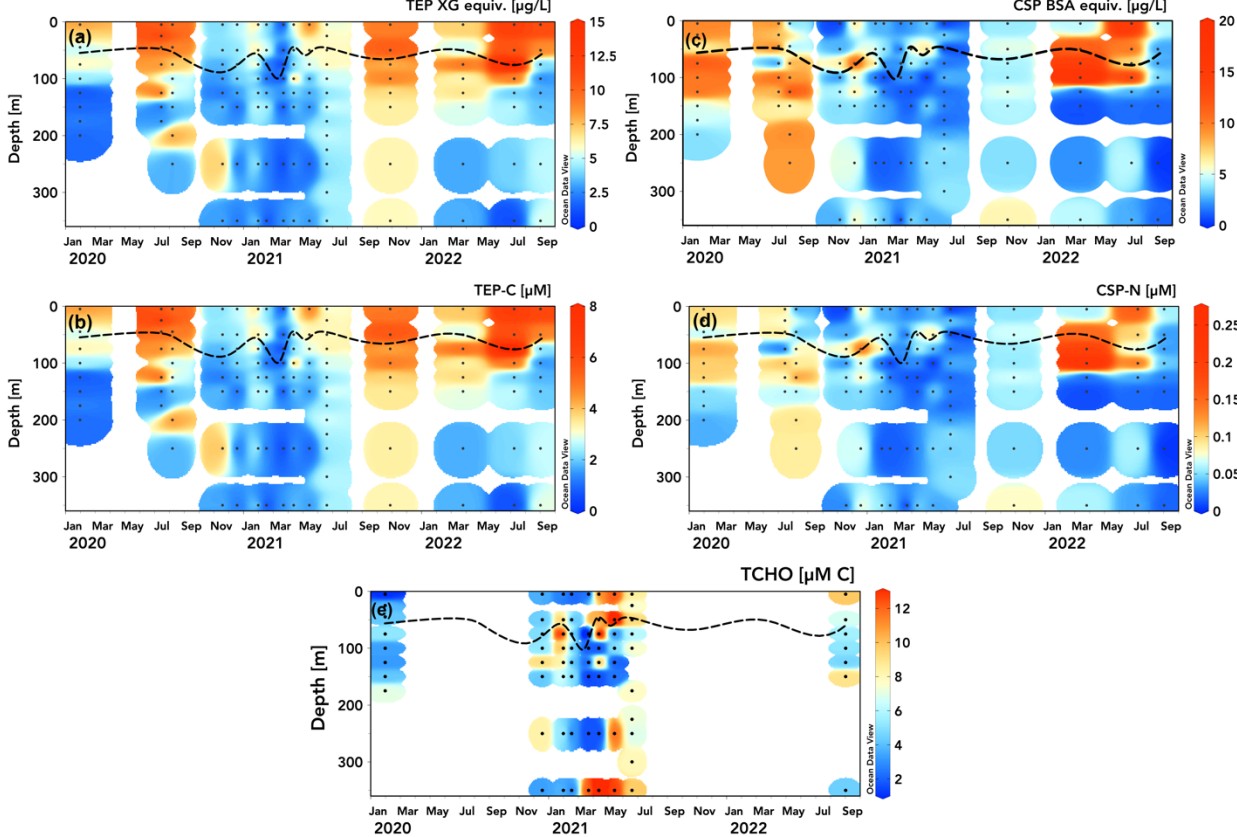

Figure 2. Station ALOHA time series (2020 – 2022) of TEP and CSP concentrations measured in xanthan gum (XG) (a) and bovine serum albumin (BSA) µg equivalents per litre (c) and converted to µM C (b) and µM N (d). Dissolved total carbohydrates (TCHO) concentrations [µM C] measured on select cruises presented in (e). Dashed line shows mixed layer depth calculated from HOT CTD data as 0.125°C decrease in temperature from the 10 m value.

At station ALOHA, TEP concentrations were highest during the summer months where values peaked within the surface mixed layer (8 – 15 XG equiv µg L$^{-1}$ (Fig. 2a); 4 – 8 µM C (Fig. 2b)), with decreasing TEP concentrations below to underlying mesopelagic waters (1 – 5 XG equiv µg L$^{-1}$ (Fig. 2a); 0.5 – 3 µM C (Fig. 2b). TEP concentrations were generally lower (2 – 7 XG equiv µg L$^{-1}$; 1 – 4 µM C), with less pronounced vertical gradients during winter months, suggesting either export of accumulated TEP from surface waters or a background of non-seasonal production or abiotic formation in deeper waters. Interannual variation in TEP concentrations in the upper 300 m is approximately 15 – 40% (coefficient of variation), with May – July 2021 having lower concentrations than similar periods in 2020 and 2022. March 2021 exhibited the lowest upper 100 m concentrations (coinciding with deepening of the surface mixed layer to 110 m after a series of storms and heavy rainfall).

The CSP distribution at station ALOHA exhibited a less observable seasonal pattern and less distinct vertical gradients as compared to TEP (Figure 2c, 2d). Elevated CSP concentrations appear to be distributed differently than TEP with high concentrations (6 – 18 BA equiv µg L$^{-1}$ (Fig. 2c); 0.1 – 0.2 µM N (Fig. 2d)) found below the surface mixed layer (50 – 100 m) and around the top of the subsurface chlorophyll max (100-125 m), consistent with the general distributions measured by Cisternas-Novoa et al. (2015) for the Sargasso Sea. CSP in 2021 was 2 – 8 BA equiv µg L$^{-1}$; 0.01 – 0.07 µM N throughout the

upper 300 m, similar to subsurface chlorophyll max and mesopelagic (>125 m) CSP concentrations in 2020
and 2022, lacking an upper ocean seasonal peak (Fig. 2c, 2d). As with TEP, CSP concentrations were
observed to be greater in March 2022 than post-storms in March 2021.
Total carbohydrate (TCHO) samples were taken on fewer HOT cruises than TEP and CSP samples during
2020-2022 due to logistical constraints. Total dissolved carbohydrates serve as a precursor substrate for
the abiotic assembly of exopolymer particles in situ (Verdugo et al., 2004). TCHO concentrations varied
between ~2 – 12 µM C across depths and season (Fig. 2e). There is a marked difference in the distribution
of TCHO concentrations between winter samples in 2020 and 2021 where surface concentrations were
low (2 – 6 µM) and data from spring 2021, where concentrations are consistently high at 350 m and in the
upper 50 m from April through June (> 10 µM). Compared to DOC measurements taken at station ALOHA,
this spring maximum at 350 m seems erroneous, but falls within the intra-annual variability of DOC at 350
m at ALOHA (± 6 µM C) and monthly variation in particulate export (Karl et al., 2021). It may be possible
that some hydrolysable particulate polysaccharides are drawn through combusted GF/F filters (Nagata et
al., 2021). Another potential explanation is the degradation and/or solubilization of exopolymers below
the subsurface chlorophyll max where polysaccharide-specific enzyme activity is elevated (Reintjes et al.,
254 2020).


3.3 Climatologies of TEP-C and CSP-N with Particulate Carbon and Nitrogen

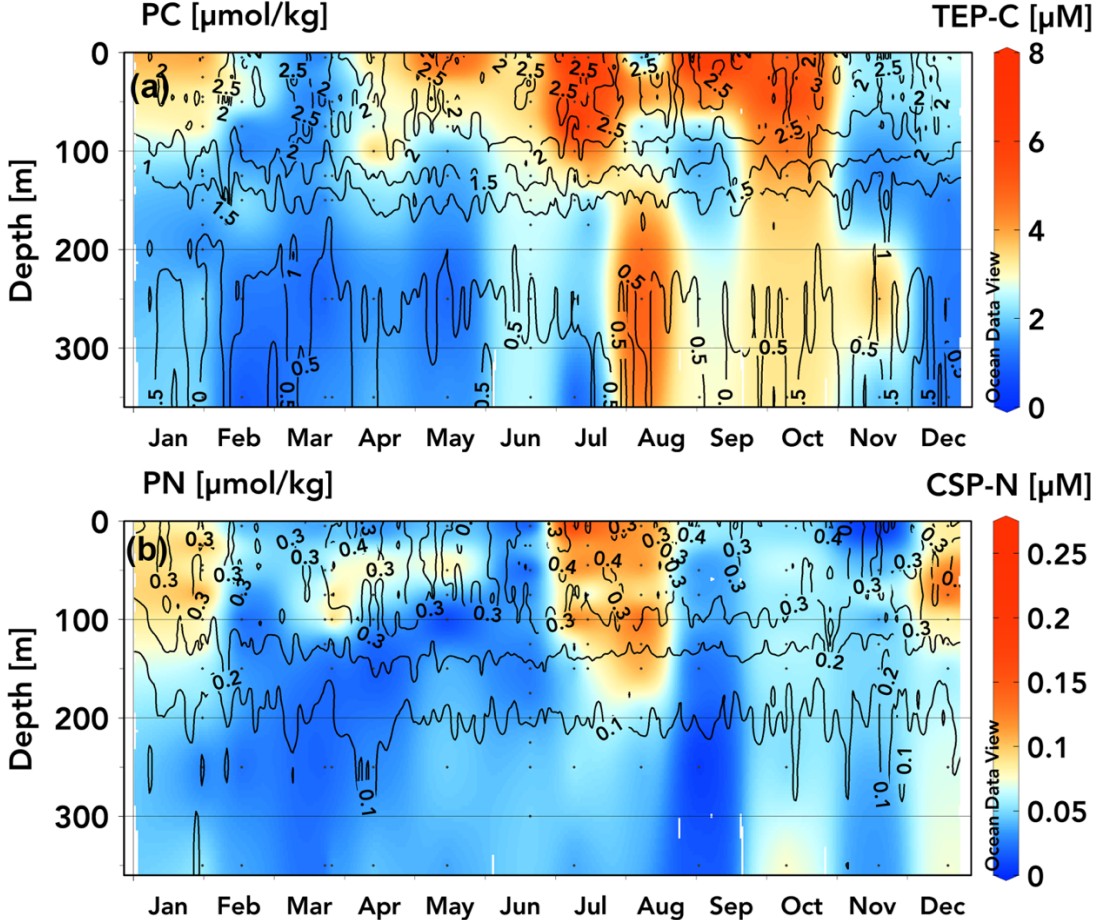


Figure 3. TEP-C (a) and CSP-N (b) (µM) concentration climatologies for 2020-2022 data measured at station ALOHA overlayed with contours from climatologies of particulate carbon (a) and particulate nitrogen (b) (µmol/kg) from the Hawaiian Ocean Time-series dataset (1989-2020 data).

Previous analyses of the climatology of upper ocean positive $preNO_3^-$ and subsurface negative $preNO_3^-$ anomaly generation at station ALOHA have revealed repeatable seasonal ingrowths of the respective anomalies during the months of April through November (Letscher and Villareal, 2018). These seasonal ingrowths are a common feature of these anomaly generation patterns across the northern hemisphere subtropics observed in the BGC-Argo float record (Smyth and Letscher, 2023). To explore exopolymer accumulation and vertical export as an explanation for these seasonal $preNO_3^-$ anomaly generation patterns, we converted the 2020-2022 TEP and CSP data to a monthly averaged climatology for station ALOHA (Figure 3), with the caveat that some months were only sampled once over the observational period. TEP exhibits a seasonal pattern with elevated concentrations found in the upper 100 m beginning in April/May (3 – 4 µM) increasing to an annual maximum in late June through early October (5 – 8 µM), followed by a decrease towards an annual minimum in February/March (1 – 2 µM) (Fig. 3a). TEP concentrations below 100 m are ~1 – 2 µM from December through June, increasing to 2 – 4 µM from June through November, concurrent with the seasonal maxima in upper 100 m TEP. We speculate that these moderate concentrations of TEP below 100 m present during summer/autumn may be due to slowly sinking aggregates as TEP accumulates in the upper 100 m through spring-summer and form aggregates before sinking, consistent with the contemporaneous peak in particulate export rates of ~30-55 mg C m$^{-2}$ d$^{-1}$ at station ALOHA (Emerson et al., 1997; Karl et al., 2012; Böttjer et al., 2017; Karl et al., 2021). The CSP climatology suggests two seasonal concentration maxima in the upper 100 – 130 m occurring in July/August and in December/January (0.07 – 0.13 µM) (Fig. 3b). CSP concentrations in other months and below these depths are <0.06 µM.

3.4 Patterns of TEP and CSP with respect to particulate C and N at station ALOHA

Comparing TEP and CSP concentrations to climatologies of particulate carbon (PC) and nitrogen (PN) respectively at station ALOHA (1989-2020; isolines in Fig. 3a, 3b) it is apparent that measured TEP concentrations reflect variation in euphotic PC more closely than CSP does PN, particularly for samples taken May-October. Elevated CSP-N concentrations during summer months (0.12-0.24 µM N) correspond with PN maxima, but during winter and spring, CSP-N comprises a smaller proportion of PN.

While CSP-N concentrations are lower in magnitude to PN concentrations, TEP-C is frequently observed to exceed background PC concentrations at station ALOHA, which may be an artifact of filtrations for PC and PN analysis losing exopolymers during GF/F filtration or excess dye binding to particles when using the colorimetric method of measuring TEP and CSP (Passow, 2002b; Bar-Zeev et al., 2011; Annane et al., 2015; Ortega-Retuerta et al., 2019; Nagata et al., 2021). The difference in nominal pore size between GF/F filters used to sample PC (0.7 µm) and the 0.4 µm pore-size polycarbonate filters used for TEP may also lead to sampling errors when comparing TEP-C and PC/POC, as most of these particles are small (<3 µm diameter) particularly in the upper 200 m, with particles tending larger as they age or sink and aggregate through the mesopelagic (Engel et al., 2020). It is therefore likely that TEP-C to PC ratios vary with depth and are more accurate for samples containing larger particles. Strands of microgels and larger particles may be easily pulled through GF/F filters under vacuum pressure and may be disaggregated when sampled in standard sediment catching methodology due to turbulence, break up at saline density layer, solubilization or rapid remineralization or preferential consumption by motile organisms (Smith et al., 1992; Buesseler et al., 2007; Fawcett et al., 2018). In addition to the variable size spectrum of TEP particles,

the electrochemistry that allows the aggregation of polymers into micro and macroscopic gels (principally
divalent cations $Ca^{2+}$ and $Mg^{2+}$) may be affected during filtration, and to a different degree with various
polycarbonate and GF/F filters (Chin et al., 1998; Meers et al., 2006). If this is the case, then gels >0.4 μm
that would otherwise be retained may be broken apart into constituent polymers or smaller nanogels that
can pass through the filter. This would lead to TEP being quantified in the DOM fraction and lead to an
overestimate of dissolved to particulate fractions of organic matter.


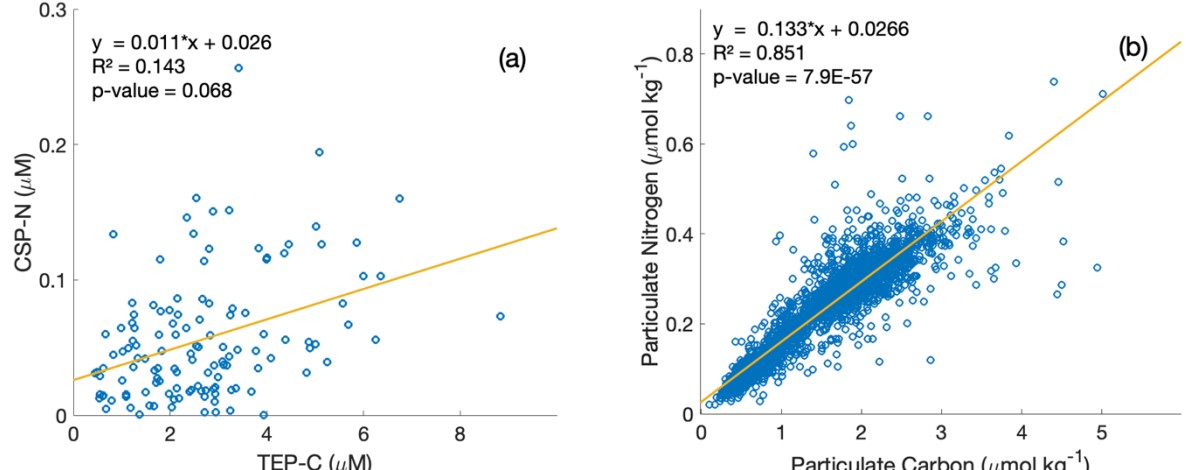

Figure 4. Property-property plots of measured CSP-N and TEP-C concentrations [μM] within the upper 350 m at
station ALOHA from this study (a) and particulate nitrogen to particulate carbon [μmol kg$^{-1}$] from the HOT dataset
(1989-2020 data) (b), including model II linear regression lines and correlation statistics.
We further explore the relationships between the exopolymer particle fraction, PC, and PN, and their
respective stoichiometries with property-property plots (Fig. 4). PN and PC concentrations are well
correlated ($R^2$ = 0.851, Model II regression) at station ALOHA (Fig. 4b) and the mean C:N ratio computed
from the inverse of the slope (7.55) is slightly higher than the canonical Redfield ratio (6.63). In contrast,
TEP-C and CSP-N concentrations show a weaker correlation ($R^2$ = 0.143, Model II regression), with an
empirical estimate of the exopolymer particle C:N stoichiometry of 90.9 or 37.3 when the regression slope
is forced through zero, the latter similar to the C:N stoichiometry determined directly on collected
exopolymers (Table 1). The weaker correlation of TEP-C with CSP-N concentrations, due to the larger
variability of the data, suggests different formation, consumption, and/or export dynamics for each group
of exopolymers, consistent with the observations of Cisternas-Novoa et al. (2015) and Zamanillo et al.
323    (2021).


## 3.5 TEP and rates of primary production

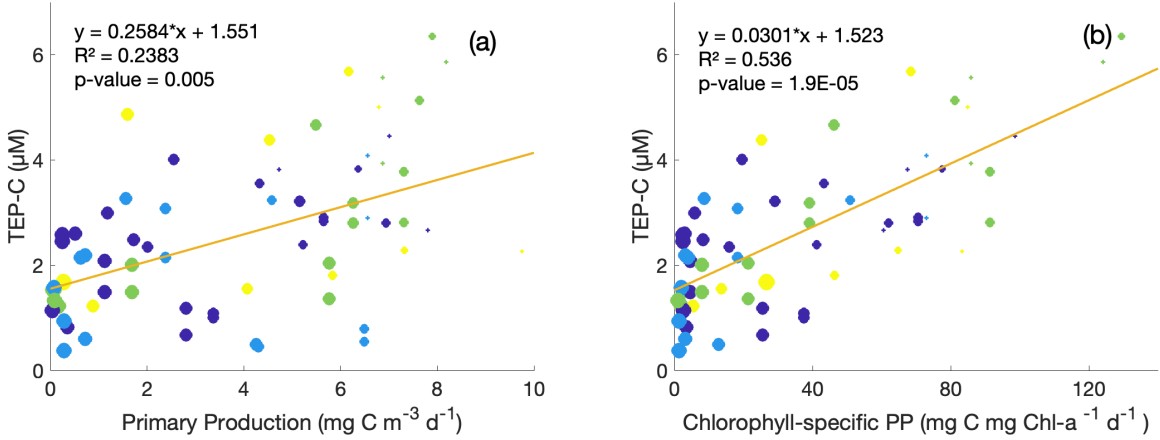

Figure 5. Property-property plots of measured TEP-C concentrations [µM C] within the upper 350 m from this study
against same-depth daily primary production (a) and chlorophyll-specific primary production (b) measured at
station ALOHA. Circle size indicates depth: larger circles are deeper samples. Colour denotes season: Purple =
winter, Blue = Spring, Green = Summer, Yellow = Autumn. Model II linear regression lines and correlation statistics
are provided.
As TEP production, abiotic formation, and consumption / degradation dynamics are often attributed to
phytoplankton community structure and downwelling irradiance intensity (Zamanillo et al., 2019; Bar-
Zeev et al., 2011; Ortega-Retuerta et al., 2009a; Berman-Frank et al., 2007; Passow, 2002a), daily primary
production (PP) measurements taken during HOT cruises were compared with TEP concentrations,
indicating a weak positive correlation for overall PP ($R^2$ = 0.24, Fig. 5a) and a stronger correlation for
chlorophyll-normalized PP ($R^2$ = 0.54, Fig. 5b). The co-occurrence of higher TEP-C concentrations and high
chlorophyll-specific primary production values in surface waters despite nutrient limitation may be
indicative of enhanced release of TEP carbohydrate precursors in addition to downregulation of
photosynthetic pigment synthesis in light-saturated surface waters (Rabouille et al., 2017; Thompson et
al., 2018). The highest values of primary productivity and TEP concentration (>4 µM) were observed in
Summer and Fall samples. There are too few data to determine whether TEP-C to PP ratios vary with
season (coloured circles, Fig. 5). CSP-N showed no such correlations with primary production within this
dataset. Although these results may be expected simply from the vertical gradients observed in TEP at
station ALOHA, chlorophyll-normalized PP gives some information on whether TEP concentrations are
only associated with surface accumulation or around the peak in chlorophyll at the subsurface chlorophyll
max. While this small dataset of TEP and PP matchups may indicate TEP production is occurring around
the subsurface chlorophyll max owing to moderate chlorophyll-normalized PP and TEP concentrations at
these depths, there are too few data at present to draw firm conclusions, particularly for near-surface
water. Wurl et al. (2011) found a similar disconnect between microbial activity and exopolymer
distributions: variations in measured TEP production rates across different Pacific waters (including late-
summer samples from station ALOHA) were not associated with phytoplankton blooms, changes in
chlorophyll concentrations or fluorescence, with abiotic formation of TEP easily maintaining observed
concentrations in the surface mixed layer (8-12 µM C $L^{-1}$ $d^{-1}$).


3.6 TEP, CSP, and TCHO concentrations on June transect 22.75°N to 31°N

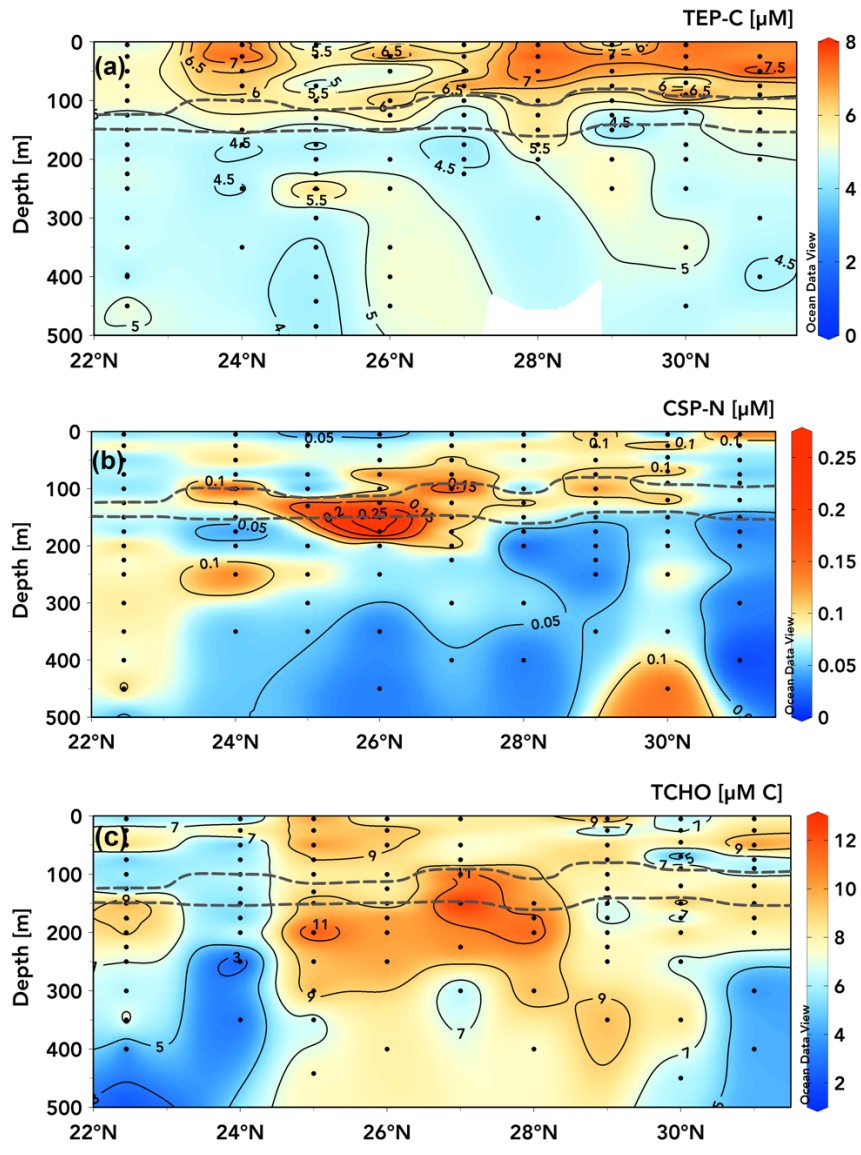


Figure 6. TEP-C [µM C] (a), CSP-N [µM N] (b), and TCHO [µM C] concentrations (with contour lines shown) within the upper 500 m measured during the June 2021 transect from 22.75°N to 31°N along ~158ºW. Dashed gray lines bound elevated CTD-mounted fluorescence, indicating the deep chlorophyll maximum.

The meridional transects of TEP-C and CSP-N concentrations in the upper 500 m taken between 22.75°N and 31°N along ~158ºW (Fig. 6) during June 2021 show an increase in upper ocean (above the subsurface chlorophyll max (gray lines, Fig. 6) TEP concentrations towards the gyre centre (Fig. 6a), with two occupations of station ALOHA at the beginning and end of the transect, separated by ten days having the lowest integrated TEP concentrations and degree of vertical gradient. From 24-31°N all stations exhibited pronounced vertical gradients in TEP concentrations between the surface and below the subsurface chlorophyll max on the order of ~2-3 µM TEP-C. At stations 24ºN and 28°N, moderate TEP-C concentrations (5.5 – 7 µM) extended into the subsurface chlorophyll max whereas high concentrations (6 – 8 µM) were restricted to the upper 75 m at 29°N and 31°N. It is unclear whether these patterns are

attributable to variable TEP production and consumption rates across depths, the production of buoyant
TEP that is retained in the surface ocean, surface turbulence and wind forcing either converging TEP to
certain latitudes or variably acting to break apart TEP particles, or gradients in phytoplankton nutrient or
oxidative stress, or photoacclimation responses affecting exudate production between sites (Sun et al.,
2018; Prairie et al., 2019). It is also of note that most stations exhibited a local increase in TEP-C at ~10-20
m immediately above the top of the subsurface chlorophyll max.  The surface maxima in TEP-C present at
28-31ºN was ~8 μM (Fig. 6a), similar in magnitude to the surface maxima accumulating seasonally in the
station ALOHA time-series (Fig. 2a, 2b; 3a). However, the vertical TEP-C gradients encountered from 24-
31ºN in June 2021 were ~2-3 μM, approximately half that observed seasonally at station ALOHA (Fig. 2b).
The observed ~2 μM latitudinal gradient in 0-100 m TEP-C concentrations (e.g. ~5 μM to ~7 μM at 22.75º
to 24ºN and similarly at 25º to 28 – 31ºN; Fig. 6a) may also be attributed to the build-up of less labile or
less export-prone (or coagulation efficient) TEP as waters move towards the gyre interior (Mari et al.,
2007; Rochelle-Newall et al., 2010; Mari et al., 2017), a feature that is also observed for the marine DOC
pool (Hansell et al., 2009).
The meridional and vertical gradients in CSP-N concentrations (Fig. 6b) throughout the transect did not
correspond to those of TEP-C (Fig. 6a). CSP-N concentrations were highest (0.15-0.26 μM) between 75-
200 m for stations 24-29°N. Profiles at 22.75°N, 30ºN and 31ºN were more uniform with moderate CSP-N
concentrations (0.05 – 0.12 μM) observed below 250 m. Elevated CSP concentrations appear to be more
closely associated with peak fluorescence signals within the subsurface chlorophyll max (gray lines) while
TEP is most abundant in the surface waters above. This disconnect between TEP and CSP distributions
suggests different dynamics in formation, residence time and decomposition and export process between
the two classes of exopolymers (Grossart et al., 2006; Engel et al., 2015; Thornton, 2018).
Measured concentrations of total dissolved carbohydrates (TCHO) varied between ~2.5-13 μM C across
the June 2021 transect (Fig. 6c). Elevated TCHO concentrations did show some overlap with elevated
surface TEP concentrations but were not consistent with TEP concentrations below the subsurface
chlorophyll max (Fig. 6c). At many stations, e.g. 25º – 28ºN, TCHO concentrations were elevated (9-12.5
μM) around and below the subsurface chlorophyll max, where TEP-C concentrations were low (4.5-5.5
μM C) but CSP-N concentrations were elevated (>0.015 μM N). Most stations exhibited vertical gradients
between surface or subsurface chlorophyll max maxima (≥10 μM) and reduced TCHO concentrations (2
– 8 μM) below 250 m, but some stations were more consistent with depth (24°N, 29°N) with peak values
near the surface. These patterns are generally dissimilar to TEP distributions that are elevated in surface
waters. Measurements from this study were on the lower end of marine dissolved carbohydrate
measurements, but consistent with previous measurements taken within the subtropical North Pacific
(Pakulski and Benner, 1994), 30-60% lower than observed across the subtropical Atlantic (Burney et al.,
1979; Goldberg et al., 2010) and 40-100% higher than in the Bay of Bengal and Arabian Sea (Bhosle et al.,
407   1998).


# 4 Discussion

## 4.1 TEP, CSP, TCHO, and exopolymer C:N stoichiometry patterns at station ALOHA

A major motivation for this study of exopolymer particle dynamics at station ALOHA was to assess the potential for this organic matter pool to help explain the shallow subsurface negative, and euphotic zone positive, $preNO_3^-$ anomalies and DIC drawdown at this site, which have thus far evaded complete accounting in the relevant tracer budgets of carbon, oxygen, and nutrients (Johnson et al., 2010; Letscher and Villareal, 2018). DIC drawdown from the surface mixed layer at station ALOHA occurs from ~April through October (Keeling et al., 2004), a period of the seasonal cycle that coincides with positive $preNO_3^-$ anomaly generation in the lower euphotic zone (~40-100 m) and negative $preNO_3^-$ anomaly generation below (~100-180 m) that are in approximate stoichiometric balance (Letscher and Villareal, 2018). These $preNO_3^-$ anomalies suggest biogeochemical processes acting on the oxygen and nitrate pools that produce $O_2$ with little nitrate drawdown in the lower euphotic zone and oxygen consumption with little accumulation of nitrate below in the shallow mesopelagic. Surface ocean production/accumulation of exopolymer particles and their export and subsequent remineralization below the shallow mesopelagic is one candidate biogeochemical process that could help explain the $preNO_3^-$ anomaly generation as well as surface mixed layer DIC drawdown in the absence of nutrient injection. This would be supported if seasonal patterns of these exopolymer dynamics match the April through October timing of peak DIC and $preNO_3^-$ changes and if the exopolymer particle pool exhibits elevated C:N stoichiometry which drive comparatively larger changes in the DIC and $O_2$ pools than nitrate.

We observe a seasonal pattern for TEP concentrations at station ALOHA that includes elevated concentrations (~4-8 µM C) above ~100 m in the late spring through early fall months, with lower concentrations (~1-3 µM C) at these depths in November through March, especially in 2020-2021 (Fig. 2a, 2b), which are similar to TEP concentrations at ~100-350 m throughout the year. CSP concentrations are generally similar in pattern, however there are notable differences such as slightly deeper upper ocean maxima (e.g. from the surface down to ~120 m) and earlier seasonal peaks in the late winter early spring and seasonal lows (ending by Sep) in the upper 100 m (Fig. 2c, 2d). The limited data on the exopolymer precursor pool of dissolved TCHO precludes a complete description of the seasonal cycle, however the most elevated concentrations (~8-12 µM C) in the upper ~100 m found in early 2021 immediately precede the growth of the seasonal peak in TEP that year beginning in May (Fig. 2e, 2b). Viewed as a monthly climatology (Fig. 3a), TEP concentrations in the upper 100 m are at the annual minimum in November through March (< 3 µM C), begin to increase in concentration from April to July, and are maintained at elevated concentrations of ~4-8 µM C through October. TEP concentrations below 100 m are low (<3 µM C) year-round with the exception of some intermediate concentrations (~3-5 µM C) in the ~100-350 m depth range in July through October (Fig. 3a), more present in 2020 and 2021 than in 2022 (Fig. 2b). The seasonal upper 100 m peak in CSP is restricted to July and August in the monthly climatology (Fig. 3b) with a wintertime peak also observed (Dec – Jan), more present in 2020 than 2021 or 2022 (Fig. 2d). The empirically estimated C:N stoichiometry of the exopolymer pool in the upper 125 m is ~1.4-2.1 times more C-rich/N-poor in June 2021 than October 2021 (Table 1), suggesting that the seasonal cycles of TEP and CSP may drive a seasonal cycle in exopolymer particle stoichiometry with the most N-poor material found during summer months. All of the above seasonal patterns in euphotic zone and upper mesopelagic TEP, CSP, and exopolymer particle stoichiometry are consistent with these pools potentially helping explain the April through October patterns of surface mixed layer

DIC drawdown, euphotic zone positive $preNO_3^-$, and subsurface negative $preNO_3^-$ anomalies at station
ALOHA.

## 4.2 Contribution of TEP production to net community production and $PreNO_3^-$ anomalies

Here we use the seasonal study of TEP distributions from the upper 350 m at station ALOHA to quantify
its potential contribution to help explain the dual enigmas of significant net community production (DIC
drawdown) from the surface mixed layer in the absence of large vertical nutrient inputs and the
generation of $preNO_3^-$ anomalies within and immediately below the euphotic zone of the subtropical
North Pacific. The potential contribution of TEP to surface excess DIC drawdown and subsurface negative
$preNO_3^-$ anomalies under nutrient limitation has been previously identified through field and lab
observations (Mari et al., 2017; Fawcett et al., 2018; Letscher and Villareal, 2018; Nagata et al., 2021). The
seasonal ~4-6 µM TEP-C concentration gradient observed between Apr-Oct in the upper 100 m and the
waters below in this study at station ALOHA, may account for a significant contribution of TEP/exopolymer
particles to both the seasonal mixed layer net community production and upper ocean $preNO_3^-$ anomalies
through the processes of TEP production, sinking or matter exported during winter mixing, and
subsequent remineralization at depth. TEP may have a significant role in exporting low-N organic matter
to underlying waters, particularly during the summer to early autumn months (Fig. 3) when the seasonal
maximum in upper 100 m TEP concentrations extends vertically into the $100 - 300$ m layer, suggestive of
vertical sinking.
The 4-6 µM vertical TEP gradient that arises seasonally at station ALOHA , e.g. 5-8 µM in the upper 100 m
Apr-Oct decreasing to 1-2 µM below, is higher than that observed by Cisternas-Novoa et al (2015)
(~10 µg XG equiv µg $L^{-1}$ / ~0.5 µM TEP-C ) in the Sargasso Sea and Wurl et al. (2011) (1.4-3.2 µM TEP-C
with one high-TEP station with a gradient of 27 µM TEP-C) in the subtropical North Pacific when applying
the carbon-converted units measured in this study. The wintertime erasure in vertical TEP gradients
between the surface and 200 m is observed in Feb-Mar and Nov-Dec samples from station ALOHA in this
study (Fig. 3a), supporting the hypothesis of TEP-C export to depth of ~100 m at ALOHA and possibly
deeper at latitudes further north in the subtropical North Pacific, via the seasonal mixed layer pump which
can deliver suspended particulate organic carbon (Dall'Olmo et al., 2016) and DOC (Hansell and Carlson,
2001) to sub-euphotic depths. This may operate in conjunction with the formation of sinking TEP
aggregates which may occur year-round (Mari et al., 2017). As its C:N stoichiometry at station ALOHA was
40-110% greater for summer than autumn, it seems that exported exopolymer particles from the surface
mixed layer to depths below may contribute disproportionally to positive and negative $preNO_3^-$ anomaly
generation during the summer months at elevated C:N stoichiometry, meaning respiration associated
with sinking exopolymers may have variable $O_2$ drawdown to nitrate release throughout the year.
The background particulate carbon flux at 150 m measured at station ALOHA of $27.8 \pm 9.7$ mg C $m^2$ $d^{-1}$
($845 \pm 295$ mmol C $m^{-2}$ $yr^{-1}$; Karl et al. 2021) would seem to indicate that the export of even a portion of
the 0-150 m integrated $750 \pm 150$ mmol C $m^{-2}$ summer/fall TEP stock by either TEP sinking or vertical
export following winter mixing would be a significant flux of carbon on an annual scale. Furthermore,
sediment trap data indicate that particulate matter exported at station ALOHA is typically slightly above
Redfieldian C:N proportions, e.g. ~8.0 (Hannides et al., 2009), while TEP measured in this study varied
between 16.4 in October to 34.3 in June (Table 1). The annual net community production rate estimated
from the seasonal DIC cycle within the surface mixed layer (~50 m) at Station ALOHA is $2.3 \pm 0.8$ mol $m^{-2}$
$y^{-1}$ (Keeling et al., 2004), thus the annual production of a surface accumulated TEP-C stock of 0.2-0.3 mol
m$^{-2}$ in the upper 50 m (e.g., $\Delta$TEP = 4 − 6 mmol C m$^{-3}$ multiplied by 50 m) may contribute 6.5-20% of the
overall net community production estimated from DIC drawdown (e.g., 0.2-0.3 divided by 2.3 ± 0.8 mol C
m$^{-2}$ y$^{-1}$), if this material is exported below. The minimum contribution of TEP to mixed layer net community
production is slightly lower, ~3.5% if the seasonal $\Delta$TEP value of ~2 mmol C m$^{-3}$ from 2020 is used.
From the calculation above, TEP production within and subsequent export below the surface mixed layer
may explain up to 20% of the total net community production, but how does this estimate compare to
the estimates of 'excess' DIC drawdown, that is DIC drawdown in excess of known N inputs (Johnson et
al., 2010), at this site? For this calculation, it is helpful to compute the N demand required to produce the
observed net community production rate, partitioned amongst the relative proportions explained by the
production of POM and DOM. Johnson et al. (2010) computed a total N demand of 287 mmol N m$^{-2}$ y$^{-1}$ at
station ALOHA assuming total organic matter production followed a C:N stoichiometry of 8.0, matching
the sinking POM stoichiometry (Hannides et al., 2009). Letscher & Villareal (2018) empirically determined
the fraction of net community production partitioned to DOM at station ALOHA from tracer budgets in
upper mesopelagic isopycnal layers from the station ALOHA climatology, finding that ~50% of net
community production is exported as DOM. We have computed the mean DOM C:N stoichiometry in the
upper 200 m at 15.5 ± 1.3 from the same climatology. Assuming net community production is partitioned
50/50% between POM and DOM with C:N stoichiometries of 8.0 and 15.5 respectively, we compute a
revised N demand of 218 mmol N m$^{-2}$ y$^{-1}$ to satisfy the observed 2.3 mol C m$^{-2}$ y$^{-1}$ net community
production within the mixed layer (Keeling et al., 2004) for the scenario whereby only POM and DOM
contribute to export production. Johnson et al. (2010) summarized total N supply to the mixed layer at
station ALOHA finding a magnitude of 144 − 201 mmol N m$^{-2}$ y$^{-1}$. Thus approximately 8 − 34% (mean =
21%) of the observed net community production N requirement is not accounted for by the known N
supply (i.e. 'unexplained') for the scenario whereby only POM and DOM export contribute to net
community production. Our study suggests that exopolymer particles may contribute as a third organic
matter pool that can be exported to balance net community production. Our estimate of TEP production
and its contribution to net community production at this site is 6.5 − 20%, with an observationally
determined C:N stoichiometry of 16.4 − 34.3 (Table 1). Addition of TEP into the surface mixed layer net
community production budget yields an N demand to explain TEP production of 4 − 28 mmol N m$^{-2}$ y$^{-1}$,
which reduces the revised total N demand of 218 mmol N m$^{-2}$ y$^{-1}$ (after accounting for elevated C:N DOM)
even further downwards to 174 − 208 mmol N m$^{-2}$ y$^{-1}$. Comparing this N demand to the prior calculated N
demand that included POM and DOM but ignored TEP, TEP contributions to the upper ocean net
community production budget help explain ~57% of the 'unexplained' excess DIC drawdown from the
surface mixed layer, i.e. by reducing the overall unexplained drawdown from a mean of ~21% to ~9%,
estimated from comparing the revised N demand for an upper ocean ecosystem including TEP production
of 174-208 mmol N m$^{-2}$ y$^{-1}$ to the estimated N supply of 144-201 mmol N m$^{-2}$ y$^{-1}$ at station ALOHA (Johnson
et al., 2010).
Table 2. Estimates of the nitrogen demand partitioned amongst POM, DOM, and TEP required to satisfy each
fractional contribution ($^{f}$NCP) of the mixed layer 2.3 mol m$^{-2}$ y$^{-1}$ net community production at station ALOHA using
their respective C:N stoichiometries. Total N supply is taken from Johnson et al. (2010) and includes vertical NO$_3$
fluxes plus N$_2$ fixation. $^{f}$NCP$_{POM}$ varies as the particulate fraction not attributable to TEP or DOM ($^{f}$NCP$_{POM}$ = 1-( $^{f}$NCP$_{TEP}$
+ $^{f}$NCP$_{DOM}$)), POM C:N from Hannides et al. (2009), $^{f}$NCP$_{DOM}$ from Letscher and Villareal (2018), DOM C:N from the
upper 200 m average of the station ALOHA climatology, $^{f}$NCP$_{TEP}$ and TEP C:N (Table 1) from this study. N demand
computed from 2.3 mol C m$^{-2}$ yr$^{-1}$ divided by C:N multiplied by $^{f}$NCP.

| Depth integration | | $^f$NCP | C:N | N demand (mmol N m$^{-2}$ yr$^{-1}$) | % of N demand |
|---|---|---|---|---|---|
| | POM | 0.30-0.435 | 8 | 86-125 | 36-52 |
| | DOM | 0.5 | 15.5 | 74 | 31 |
| 50m | TEP | 0.065-0.20 | 16.4-34.3 | 4-28 | 2-12 |
| | **Total demand** | | | **174-208** | |
| | **Total supply** | | | **144-201** | |


Lastly, we compare the seasonal TEP cycle observed at station ALOHA from 2020-2022 to previous
estimates of the formation rates of residual preNO$_3^-$ anomalies within and immediately below the
euphotic zone. Letscher and Villareal (2018) estimated the seasonal (~Apr-Oct) development of a residual
positive preNO$_3^-$ anomaly (i.e. the residual anomaly after accounting for non-Redfield POM and DOM
stoichiometry) within the upper 100 m with a climatological magnitude of 0.53 ± 0.27 µM N. A similar
seasonal negative preNO$_3^-$ anomaly develops between ~100-180 m with a climatological magnitude of -
0.54 ± 0.25 µM N over a ~180-day period from Apr-Oct, consistent with surface TEP accumulation before
winter mixing (Fig. 2). With an assumed 1:1 C:O$_2$ stoichiometry of TEP formation and remineralization (as
for nearly pure carbohydrate material), the consumption of 4-6 µM seasonally exported TEP C (at a C:N
ratio of 25 ± 8) should release the equivalent of 0.12-0.35 µM nitrate which is 23-67% of the 0.53 µM
mean residual negative preNO$_3^-$ anomaly and 22-64% of the 0.54 µM mean residual positive preNO$_3^-$
anomaly. These values for TEP's potential contribution to preNO$_3^-$ anomalies assume the export of surface
TEP to underlying waters 100-200 m where they are subsequently remineralized. If a large proportion of
seasonal TEP production is quickly exported to the deeper mesopelagic through aggregation and
gravitational settling or winter mixing, then these values will likely be overestimates. Remaining
mechanisms to explain the remainder of preNO$_3^-$ anomaly formation include mining of sub-euphotic zone
nitrate by vertically migrating phytoplankton (Pilskaln et al., 2005; Villareal et al., 2014) and heterotrophic
bacterial uptake of nitrate when consuming C-rich organic matter such as TEP (Fawcett et al., 2018).
Finally, we note that moderate concentrations of TEP at 150-350 m (3-5 µM C) are present throughout
the late summer to early autumn months at station ALOHA (Fig. 2b, Fig. 3a), but whether these
concentrations represent matter exported from the surface or subsurface chlorophyll max below the
depth of the negative preNO$_3^-$ anomaly (~100 – 180 m; Letscher & Villareal, 2018), or separate activity in
the upper mesopelagic is unclear. Compositional analysis of TEP molecules and polysaccharide-associated
enzymes throughout the water column and over an annual cycle may elucidate sources and sinks of TEP
beyond physical sinking and mixing processes.
## 4.3 TEP, CSP, and TCHO meridional patterns in the NPSG
Previous observations of TEP and CSP particle concentrations in high latitude oceans and temperate shelf
seas have observed that both exopolymers are coupled to chlorophyll distributions (Beauvais et al., 2003;
Busch et al., 2017; Nosaka et al., 2017; Anastasi, n.d.; von Jackowski et al., 2020). Other mid-latitude
regions such as the Sargasso Sea (Cisternas-Novoa et al., 2015) and Catalan Sea (Zamanillo et al., 2021)
exhibit different dynamics, where TEP is disconnected from CSP distributions as was observed in this study
in the subtropical North Pacific.
Exopolymers measured across the upper 500 m of the subtropical North Pacific in June 2021 (Fig. 6) were
found to have depth gradients similar to summertime conditions at station ALOHA (Fig. 2, 3), with elevated
concentrations in surface waters for TEP and around the subsurface chlorophyll max for CSP and lower
concentrations below in the shallow mesopelagic. There are also meridional gradients present (Fig. 6) with
increasing surface ocean TEP concentrations northwards towards 31ºN while CSP is most elevated at 24
– 30ºN. Without additional data on TEP/exopolymer molecular composition it is difficult to ascertain
whether this meridional gradient represents accumulation of more refractory TEP, or enhanced
production/depressed export from waters towards 31ºN. However, this TEP meridional gradient matches
that observed for surface ocean DOC in the region (Abell et al., 2000) which is thought to arise from the
convergence of a semilabile component of DOC with lifetimes of years by the Ekman circulation across
subtropical gyres (Hansell et al., 2009). The C:N stoichiometry of the exopolymer particle pool was also
found to increase from station ALOHA to 31ºN both at the surface (~26 vs. ~33) and at 125 m (~34 vs. ~38)
(Table 1). Dissolved TCHO were more elevated (> 7 μM C) north of 24ºN as well (Fig. 6c). The observed
disconnect between TEP and TCHO distributions may be attributed to both formation and degradation
processes: precursors being created around the subsurface chlorophyll max by phytoplankton and
resultant low-density TEP particles concentrating in surface waters or sinking TEP being hydrolyzed below
the subsurface chlorophyll max by bacteria, yielding reduced TCHO concentrations. The latter process is
consistent with the hypothesized remineralization of low-N organic matter requiring heterotrophic nitrate
uptake, generating a negative $preNO_3^-$ anomaly (Fawcett et al., 2018). Compositional analysis of TEP
particles, dissolved sugars, and stable isotopic measurements of the relevant nutrient and organic matter
N contents through the upper 400 m would help confirm. Lastly, a lack of spatiotemporal coherence in
the distributions of TEP and its precursor TCHO may result from differing timescales over which they are
biotically cycled, with the latter possibly processed 3-10 times faster than other common labile organic
materials like amino acids by bacteria in open ocean environments (Kaiser and Benner, 2012).
The meridional patterns observed in TEP, CSP, TCHO, and exopolymer particle C:N stoichiometry are all
suggestive that the contributions of exopolymer particle dynamics to upper ocean net community
production, export, and nutrient cycling diagnosed at station ALOHA (Sect 4.2) may play a larger role
further north towards the core of the subtropical North Pacific gyre. While the TEP concentrations
measured in this study were low (~2-15 XG equiv. μg/L; 1-8 μM C) compared to other regions (e.g. ~20-40
XG equiv. μg/L in the Sargasso Sea (Cisternas-Novoa et al., 2015); ~1-20 μM C in the tropical North Pacific
(Wurl et al., 2011)), their highly carbon-enriched stoichiometry (particularly in summer with C:N = 26-38)
means that these particles are a significant component of the upper ocean organic matter pool. The
seasonal and latitudinal variation we observed in carbon and nitrogen conversion factors suggest using a
single factor will bias many estimates of TEP-C and CSP-N from dye-binding assays. We therefore hope
that more effort will be made in future studies to constrain TEP and CSP elemental stoichiometry to
compare exopolymer concentrations from different depths, seasons and locations with greater
confidence.

## 5 Conclusions and Future Directions

The seasonal, interannual and meridional variation of TEP and CSP observed in this study reinforces the
building evidence that exopolymer production, accumulation and remineralization are not static
processes, even in oligotrophic regions (Radić et al., 2006; Cisternas-Novoa et al., 2015; Engel et al., 2015;
Zäncker et al., 2017). Further process experiments that incorporate TEP and CSP dynamics with respect to
other biological and chemical parameters are needed to understand the biogeochemistry of each
exopolymer type for a given location and season, aiding efforts to model both with respect to other
parameters through depth and time at a synoptic scale. Work that helps to validate the sources and sinks
of exopolymers within the water column is particularly important in quantifying how much carbon is
exported from or cycled within surface waters (including the surface microlayer) and where these
molecules are remineralized. Compositional analysis of TEP particle and dissolved carbohydrates
compositions and associated proteomic or transcriptomic analyses may elucidate the vertical distribution
of TEP production, enzymatic hydrolyzation and remineralization of the resulting labile monomeric sugars.
TEP concentrations measured with the Alcian blue spectrophotometric method and converted to µM C
with our empirically derived carbon conversion factors were found to be greater than GF/F collected
particulate carbon measurements from the Hawaiian Ocean Time-series. Additionally, the estimated C:N
stoichiometry of 16.4 – 38.1 for exopolymer particles from this study is significantly C-rich/N-poor relative
to the C:N of the sinking flux collected in sediment traps at station ALOHA, 8.0 (Hannides et al., 2009). This
supports the hypotheses that TEP and marine microgels may be 'missed' by traditional sampling
techniques for sinking and suspended particulate organic carbon (Quigg et al., 2021), possibly due to
disaggregation of the gel-particles upon encountering the GF/F filter or collection brine of sediment traps
as well as potential rapid microbial remineralization within trap cups (Fawcett et al., 2018 and references
therein). Future research is required to resolve the mechanisms leading to inefficient collection of TEP
within standard marine particle sampling protocols and fully integrate TEP and marine gels sampling
within marine carbon biogeochemistry studies. Further work is also needed to ascertain the degree to
which exopolymer particles are exported below the surface ocean via slow gravitational sinking and/or
vertical mixing within the seasonal mixed layer pump (Quigg et al., 2021; Mari et al., 2017).
Though TEP sinking rates, remineralization rates and C:$O_2$ respiration stoichiometry are not addressed in
this dataset, previous studies in analogous regions indicate that the summertime production of highly
non-Redfieldian exopolymers and potential winter export observed in this time series may explain a
significant portion of subtropical positive and negative preNO$_3^-$ anomalies (22-67%), consistent with this
mechanism's description and modelling by Letscher and Villareal (2018). Uncertainty in the contribution
of TEP/exopolymers to preNO$_3^-$ anomalies (and excess DIC drawdown) primarily results from variability in
the total TEP upper ocean accumulation and its C:N ratio; with some evidence for seasonal, vertical, and
meridional differences in these ratios evidenced in this study. The upper ocean exopolymer cycle helps to
close the C, N, and $O_2$ budgets at station ALOHA by contributing 8.5-20% of net community production
and reducing the 'missing' mixed layer DIC drawdown and N supply by ~57% and ~12%, respectively. While
leaving room for significant contributions from other processes such as vertically migrating phytoplankton
and heterotrophic nitrate uptake to be further validated. More frequent measurements of TEP
concentrations and its stoichiometry from the subtropical North Pacific and elsewhere would help
quantify this potentially overlooked component of the ocean's biological pump operating across the vast
subtropical gyres.

**Author contributions**
RTL and TV conceptualized this study as part of NSF grants 1923687 and 1923667 "Transparent
exopolymer and phytoplankton vertical migration as sources for preformed nitrate anomalies in the
subtropical N. Pacific Ocean". KC, RTL, and HOT technicians performed fieldwork; KC performed
laboratory analyses for TEP, CSP and TCHO and respective data analyses. KC, RTL, and TV contributed to
writing and editing. Data from the Hawaiian Ocean Time series were obtained via the Hawaii Ocean Time-
series HOT-DOGS application; University of Hawai'i at Mānoa. National Science Foundation Award #
658     1756517.

**Acknowledgements**
We would like to thank the crew and technicians aboard the RV Kilo Moana for their assistance in
collecting samples through the COVID pandemic and assisting during the June 2021 transect cruise. We
are grateful to Angelicque White (UH-Manoa) for her assistance and leadership in accommodating the
TEP, CSP, and TCHO sampling on the 2020-2022 HOT cruises and to Brandon Brenes (UH-Manoa) for much
of the sample collection at sea. We also wish to thank former UNH graduate students Jessica Gray and
Sarah Benson for their assistance with sampling during the 2021 cruise.

**Financial support**
This study was funded as part of NSF grant 1923687 to RTL and 1923667 to TV entitled: "Collaborative
research: Transparent exopolymer and phytoplankton vertical migration as sources for preformed nitrate
anomalies in the subtropical N. Pacific Ocean."

**Data availability**
The data reported in this study are available at: https://www.bco-dmo.org/project/772658.

**Competing interests**
We declare no competing interests in the undertaking and publication of this study.

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
