# Peer review of "A time series analysis of transparent exopolymer particle distributions and C:N"

_EGUsphere, 2024_

## Author Response (AR1)

Referee #1

The authors present data on TEP and CSP measurements over the course of two years at station ALOHA and a transect that headed north from station ALOHA conducted in June of 2021. The resulting data are then used to estimate the contribution of TEP to carbon cycling in the region. The presentation of the data is challenging at times as the authors skip over the presentation of some data, and assume the readers can recall the climatology of organic nitrogen and carbon at station ALOHA. There is also only limited statistical analysis of these data, which leaves this reader unclear as to significance of some of the differences highlighted by the authors. The presentation of the calculations on the relative impact of TEP on carbon cycling is also not well presented, and it was difficult to follow the assumptions made by the authors.

This manuscript uses an excessive number of abbreviations which makes it challenging to follow the primary points being made by the authors. The manuscript centers on TEP and CSP, adding in abbreviations for NPN, PPN, SCM, NCP, topics that are ancillary to the primary topic makes the abstract hard to follow. This is particularly true since the abbreviations change later in the manuscript when PreNO3 is used.

**Thank you for pointing out when our use of abbreviations detracts from the reading than improving its flow. We have removed all uses of NPN, PPN, SCM, and NCP and now use their fully expressed phrases. We have retained use of TEP, CSP, TCHO, and preNO$_3^-$.**

The abstract should focus more centrally on work presented in this manuscript. The first paragraph is an introduction to the general ideas of the inorganic carbon drawdown, and can be summarized in one sentence before moving on to measurements made in the current project.

**Thank you for this comment. We have revised the Abstract downwards in length with an eye toward brevity.**

Line 199 – here the authors note they will present data in units of umC and umN, but then they go on to present both the xanthan and BA units and carbon/nitrogen units (starting ~ line 233 and in figure 2).

**We first present the TEP and CSP data in Figure 2 using the semi-quantitative xanthan gum and bovine albumin equivalents units to be comparable with the existing literature on this topic. Since our motivation is to *quantitively* link the dynamics of the TEP and CSP pools to the observed DIC and preNO$_3^-$ anomalies at station ALOHA, we then present the TEP and CSP pools in quantitative µM C and µM N units using our empirically derived carbon conversion and nitrogen conversion factors for the region and primarily refer to these units throughout the remainder of the manuscript.**

Line 208: are the differences in the conversion factors for nitrogen and carbon between the 5 m and 125m statistically significant? What about the C:N ratios. If they are significantly different, please provide the results if the statistical analysis. If not, please do not highlight differences.

**We have performed the requested statistical tests and reported the p-values from Welch's t-tests for two samples with non-equal variances whenever we make statements in this section comparing the carbon conversion factors, nitrogen conversion factors, and resulting C:N stoichiometries across depths, latitudes, and seasons.**

Line 243 'Interannual variation in TEP concentrations...' I am unclear on what data are used for this calculation – bulk concentrations? Surface water? Water-column integrated values? Please be more specific.

**We have edited this statement at line 381 to include that the interannual variability in TEP concentrations is computed within the upper 300 meters, reporting the coefficient of variation.**

Line 247 'CSP … more variable than TEP...' based on what quantifiable metric? This conclusion appears to be based on visual inspection of the data, which is emphasized by the vague mention of differences at 'specific depths' and concentrations that 'appear to be...' different.

**We have edited this statement at line 385 to read, "The CSP distribution at station ALOHA exhibited a less observable seasonal pattern and less distinct vertical gradients as compared to TEP (Figure 2c, 2d)."**

Line 262: Given the interannual variability mentioned on the previous page, I do not think presenting an annual climatology is a valid presentation of the data. Using CSP as an example, the data in figure 2 show March has either the highest water column values or the lowest, so this averages out to a number in the middle. The comparison starting at line 274 to the climatology of PC and PN is also not shown in this manuscript and relies on the user to have a priori knowledge of PC and PN data at station ALOHA. Furthermore, presenting plots of PC vs. PN and TEP-C vs. CSP-N in figure 4 is not an effective means of discussing differences between TEP-C and PC.

**Thank you for this critique. We have added the following text to begin section 3.3 on the topic of TEP and CSP climatologies to better justify our presentation of them and discuss their caveats.**
**"Previous analyses of the climatology of upper ocean positive preNO$_3^-$ and subsurface negative preNO$_3^-$ anomaly generation at station ALOHA have revealed repeatable seasonal ingrowths of the respective anomalies during the months of April through November (Letscher and Villareal, 2018). These seasonal ingrowths are a common**

**feature of these anomaly generation patterns across the northern hemisphere subtropics observed in the BGC-Argo float record (Smyth and Letscher, 2023). To explore exopolymer accumulation and vertical export as an explanation for these seasonal preNO$_3^-$ anomaly generation patterns, we converted the 2020-2022 TEP and CSP data to a monthly averaged climatology for station ALOHA (Figure 3), with the caveat that some months were only sampled once over the observational period."**

line 382 'observed ~2 μM latitudinal gradient' – I am not clear on what is presented here as 2 um, is this an integrated water column value?

**We have edited this statement at line 667 to clarify that the ~2 μM C TEP latitudinal gradient within the upper 100 m is estimated by comparing the values of ~5 μM to ~7 μM at 22.75º and 24ºN and similarly for 25º to 28-31ºN.**

Line 492: this is the first time the authors have mentioned a depth integrated carbon value. The conversation comparing existing data with the depth-integrated TEP-C value would be far more compelling if the authors presented the month/year/cruise depth integrated values for their own data (and *not* using the climatology given my previous comment about averaging such disparate years).

**The climatological seasonal gradient in upper 100 m TEP accumulation from Figure 3a is ~4-6 μM C, which we used in subsequent calculations in Sect. 4.2 linking TEP dynamics to net community production and preNO3- anomalies. The separate seasonal gradients in 2021 and 2022 are very similar, e.g. ~4-6 μM C while the seasonal gradient in 2020 is reduces, e.g. ~2 μM C. We have added a statement at line 914 to address the interannual variability in our estimates of the potential contribution of TEP dynamics to net community production rates:**
**"The minimum contribution of TEP to mixed layer net community production is slightly lower, ~3.5% if the seasonal ∆TEP value of 2 mmol C m$^{-3}$ from 2020 is used."**

Line 501: The section linking the TEP data to net community production is difficult to follow as it jumps from one idea to the next and the reader is never given the citation link to Table 2. Further the various facts from the existing literature are presented, but there is no clear formula provided as to how the calculations are done. This may be an obvious calculation to the authors, but although I am an oceanographer, I am not versed on each step of this calculation and thus the steps used for these calculations are not clear. This is further confused by the mention of one citation (Hannides et al.) in the table legend that does not appear in the paragraph that is apparently describing this calculation.

**Thank you for this helpful comment where our arguments were not clear. This section of the manuscript (~line 912-947; including the Table 2 caption) has been edited for clarity, including which numbers are used in arithmetic, and adding more citations when appropriate.**

Line 641: although the manuscript indicates the data are available at BCO-DMO, there are no data available online at this time (October 2024).

**Thank you for pointing this out. The data have been deposited on April 4, before resubmission of this revised manuscript.**

Table 1 – since the authors talk about TEP first, it would be easier to interpret this table if the columns start with TEP from the left before proceeding on to CSP and then the C:N ratios.

**This change to Table 1 was completed.**

Figure 2 – the fonts are so small on these figures that it is quite difficult to gather the information needed to understand the figures. Even at 200% magnification, I can just barely read the axes.

**The font and axes size has been increased on all Figures 2-6.**

Figures 4 & 5 : are these model II regressions statistically significant? Please present the p-values in addition to the r2 values. The colors in figure 5 are difficult to sea, the yellow in particular.

**We have added the p-values and made the colored dot symbol sizes larger on these plots to make it easier for the reader.**

Figure 6: this figure would be easier to interpret if the latitudinal range shown on the x-axis were the same for all three subplots. The shift in axes makes it challenging to line up the three variables being plotted. The data for TEP and CSP particles would also be easier to interpret if the contour lines for fluorescence were imposed upon the color plots for TEP and CSP.

**The latitudinal range on the x-axes of both Figure 6 subplots have been made the same. The elevated fluorescence contour lines have been added to subplots a and b with subplot c removed.**

Figure 7 – this figure would be easier to interpret if the same color scheme were used on both subplots.

**The color palettes and ranges have been coordinated between all subplots within Figures 2, 3, and 6.**

I am confused as to whether or not the data from station ALOHA from figure 6 are the same data that appear in figure 2. It seems not as the transect data show TEP-C concentrations in the upper 100 meter has having carbon concentrations ~5 µM C, but figure 2 shows

values ~3 µM C. Am I missing something here as it seems odd to separate data from a transect that starts at station ALOSH from a regular sampling of station ALOHA, and such a discrepancy in the measurement is an indication of the variability in this measurement.

**Figure 2 presents the data sampled at Station ALOHA at regular intervals over the period of 2020-2022 (i.e. the x-axis is time), whereas Figure 6 presents a single transect completed from station ALOHA into the gyre and back south to ALOHA (i.e. the x-axis is latitude). Figure 6 is to show the meridional variation over this transect vs Figure 2 which shows the seasonal to interannual variability (as far as can be captured from a short, incomplete time series).**

Referee #2

Curran and co-authors present an important dataset on carbon-rich exopolymers in the North Pacific Subtropical Gyre, exploring the data to address the contribution of these compounds to net production and the associated surface nitrogen budget. This is an important contribution to the age-old question of why NCP in subtopical gyres is greater than can be accounted for by the particulate sinking flux and DOM flux.
My comments on the manuscript are largely editorial as this is a well executed study.
I found that it took a "long time" to get to the crux of the paper that contextualizes the data in reference to the carbon/nitrogen budgets of the NPSG. My recommendation is to revert to a traditional presentation where results are presented first, followed by a discussion section. As presented, the first sections of the combined Results and Discussion felt repetitive and bordered on tedious. For the short-attention-span readers we have all become, I think this structure would go a long way to better engaging readers.

**Thank you for the advice on improving the manuscript to be a more engaging and concise read. We have streamlined the Abstract and Conclusions sections as well as performed a significant restructuring of the Results and Discussion to separate them as distinct sections.**

Referee #3

The authors present exopolymer gel particle data from the ALOHA station time series and a detailed oceanographic study. This is a valuable dataset that contributes to the understanding of the estimated imbalance between nitrogen supply and demand in the North Pacific subtropical gyre. Furthermore, the authors give an estimate of the nitrogen content of CSP particles using the colorimetric method, which had not been done before. However, the structure of the manuscript and the quality of the figures make it difficult to read. Major comments:

1. Overall, the abstract and conclusion are too long, and the introduction is too short, lacking information on TEP and CSP. It should be stated that EPS acts as a bridge

between dissolved and particulate fractions of the organic matter which leads to a size continuum of particles in the ocean (see Verdugo et al., 2004; Verdugo, 2012) and that TEP and CSP are independent particle classes with different origins and fates (e.g. Cisternas–Novoa et al., 2015; Zamanillo et al., 2021). It is known that TEP influences the particle/carbon export (i.e. buoyancy, stickiness, aggregation) and therefore influences the distinction between suspended and sinking particles. However, those properties remain unknown for CSP. An overview of the knowledge on TEP and CSP can be found in Mari et al., (2017; https://doi.org/10.1016/j.pocean.2016.11.002), Thornton et al. (2017; https://doi.org/10.3389/fmars.2018.00206) and in the introduction of Marina Zamanillo Campos' thesis (2019).

**Thank you for this feedback. As stated in our response above to Reviewer #2, we have streamlined the Abstract and Conclusions sections as well as separated the Results and Discussion into separate sections. We have added text to the Introduction to discuss the points suggested by the reviewer at line 137 – 160:**

**"Exopolymers act as a bridge between the dissolved and particulate fractions of marine organic matter, with dynamic assembly and disassembly of marine gels helping to fill the size continuum of particles in the ocean (Verdugo et al., 2004; Verdugo, 2012). The related but distinct Coomassie stainable particles (CSP) are thought to track the more protein-rich component of the marine exopolymer/gel pool, which likely impacts the fate of these particles differently than the polysaccharide-rich TEP pool (Cisternas-Novoa et al., 2015; Zamanillo et al., 2021)."**

2. The results lack the statistical comparisons needed to attest to the significance of the results obtained.

   **We have added statistical tests for significance when comparing TEP and CSP carbon and nitrogen conversion factors and the resulting C:N stoichiometry from Table 1. Similarly for comparisons of CSP to TEP and PON to POC in Figure 4 and TEP concentrations against chlorophyll in Figure 5. The calculations relating TEP/exopolymer particle dynamics to surface mixed layer DIC and upper ocean preNO$_3^-$ anomaly budgets in Table 2 use ranges of tracer concentrations/fluxes to represent the inherent seasonal/interannual variability and analytical uncertainty.**

3. The paragraphs that make up the conclusion should be incorporated into the discussion section. The conclusion should give an exhaustive overview of the study's results.

   **The Conclusions section has been edited and streamlined, including adding some passages to relevant sections within the Discussion.**

4. Many sentences are too long (4-5 lines) and too many abbreviations are used. For consistency with other studies, I suggest replacing maximum subsurface chlorophyll (SCM) with deep chlorophyll maximum (DCM) and deleting the abbreviation SML, which stands for sea surface microlayer. Instead, indicate the mixed layer.

**As stated above in response to Reviewer #1, we have removed many usages of abbreviations, including SCM and DCM mentioned here by the reviewer. Many sentences throughout the manuscript have been edited during the revision process.**

5. The size of the panels in the figures should be standardized to make them easier to read (figures 2, 3 6, and 7).

**The color palettes and ranges have been coordinated between all subplots within Figures 2, 3, and 6 as well as the sizes of the subplots.**

Minor comments:
Line 83: Remove sometimes

**Change completed.**

Line 162: add a space between the number and the unit (6mL to 6 mL) and make the correction through the manuscript.

**Change completed.**

Line 251: Correct "0.01 – 0.0.07 µM" to "0.01 – 0.07 µM"

**Change completed.**

Line 290: change swimmers to "motile microorganisms"

**Change completed.**

Lines 309-310: You can add that the statement is consistent with Cisternas–Novoa et al., 2015 and Zamanillo et al., 2021 observations.

**Change completed.**

Line 451: To facilitate the comparison, figure 7 should be removed and incorporated in other figures: Panel A should be included in Figure 2 and Panel B in Figure 6

**Changes completed.**

Lines 449-452: The sentence is too long, divide it into two parts.

**Change completed.**

Lines 452-457: Confusing you can simplify by stating that here you had dissolved hydrolyzable carbohydrates which were originally different glycans with different degradation properties e.g. sulfated fucoidan (Vidal-Melgosa et al., 2021; https://doi.org/10.1038/s41467-021-21009-6) or laminarin (becker et al., 2020; https://doi.org/10.1073/pnas.1917001117)

**This sentence has been edited for clarity at line 1048 to read:**
**"Lastly, a lack of spatiotemporal coherence in the distributions of TEP and its precursor TCHO may result from differing timescales over which they are biotically cycled, with the latter possibly processed 3-10 times faster than other common labile organic materials like amino acids by bacteria in open ocean environments (Kaiser and Benner, 2012)."**

---

## Author Response (AR2)

Referee #3

The revised version of the manuscript is much clearer and easier to follow. Only minor concerns remain:

**Thank you, we are glad the revisions have improved the clarity and presentation of our study.**

-Replace bovine albumin with bovine serum albumin and the abbreviation BA with BSA.

**This change has been completed on lines 163, 221, and the Figure 2c title.**

-Figure 3: Provide a link or reference for the Hawaiian Ocean Time Series dataset.

**The link to the HOT dataset has been added on line 125.**

-In Figure 5, primary productivity and chlorophyll-specific primary production data are shown, but how primary productivity and chlorophyll a were measured is not provided in the method section.

**The PP and chl a data were measured as part of the HOT time-series and have the same source as the data from Figure 3 on particulate carbon and nitrogen. A statement has been added to the Methods section at lines 123-125 introducing where these data can be obtained including the link.**

**"Primary productivity, chlorophyll a, particulate carbon, and particulate nitrogen data measured as part of the HOT program for the period 1988-2022 were obtained from https://hahana.soest.hawaii.edu/hot/hot-dogs/."**

-In the Conclusions and Future Directions section, the paragraph "Following the conversion of semi-quantitative [...] (or missing N supply) to ~9%." Is redundant from the discussion.

**This paragraph at lines 638 to 646 has been deleted. A clause has been newly added at line 646-647 that summarizes the main point from the deleted paragraph which we seek to reiterate here in the Conclusion section as it is one of the main conclusions of the study.**

"... at station ALOHA by contributing 8.5-20% of net community production and reducing the 'missing' mixed layer DIC drawdown and N supply by ~57% and ~12%, respectively."